# Prognostic Factors of the Inability to Bear Self-Weight at Discharge in Patients with Fragility Femoral Neck Fracture: A 5-Year Retrospective Cohort Study in Thailand

**DOI:** 10.3390/ijerph19073992

**Published:** 2022-03-28

**Authors:** Paween Tangchitphisut, Jiraporn Khorana, Phichayut Phinyo, Jayanton Patumanond, Sattaya Rojanasthien, Theerachai Apivatthakakul

**Affiliations:** 1Department of Orthopaedics, School of Medicine, Mae Fah Luang University, Chiang Rai 57100, Thailand; paween.tan@mfu.ac.th; 2Department of Surgery, Faculty of Medicine, Chiang Mai University, Chiang Mai 50200, Thailand; jiraporn.kho@elearning.cmu.ac.th; 3Department of Family Medicine, Faculty of Medicine, Chiang Mai University, Chiang Mai 50200, Thailand; 4Center for Clinical Epidemiology and Clinical Statistics, Faculty of Medicine, Chiang Mai University, Chiang Mai 50200, Thailand; jpatumanond@gmail.com; 5Musculoskeletal Science and Translational Research (MSTR) Cluster, Chiang Mai University, Chiang Mai 50200, Thailand; 6Department of Orthopaedics, Faculty of Medicine, Chiang Mai University, Chiang Mai 50200, Thailand; srojanas@gmail.com (S.R.); tapivath@gmail.com (T.A.)

**Keywords:** prognostic factors, weight bear, inability, femoral neck fracture, discharge

## Abstract

An inability to bear self-weight is one of the unfavorable results in geriatric hip fracture, which needs to be prevented. This study determines pre-operative, intra-operative, and post-operative prognostic factors of the inability to bear self-weight at discharge in patients with fragility femoral neck fracture. This retrospective study was conducted at Chiang Mai University (CMU) hospital with an observational cohort design. Electronic medical records of patients aged ≥ 50 years old with fragility femoral neck fractures between 1 January 2015 and 31 December 2019 were reviewed. Pre-, intra-, and post-operative factors were collected. Ambulation status at discharge time was classified into either ability or inability to bear self-weight. Analysis of prognostic factors was done using multivariable risk ratio regression. In total, 269 patients were recruited in this study. Significantly prognostic factors of inability to bear self-weight at discharge were end-stage renal disease (ESRD), cirrhosis, cerebrovascular disease, pre-fracture ambulatory status, having associated fractures, increasing intra-operative blood loss, and having pressure sore. These prognostic factors could be used to predict patients’ outcomes at discharge. Proper management could then be offered to the patients by the multidisciplinary care team to enhance surgical outcomes.

## 1. Introduction

During the past decades, Thailand has been continuously growing into a complete aging society [1]. An increase in the proportion of the geriatric population around the globe comes with a significant healthcare burden to public health. One of the most common injuries is geriatric hip fractures. In Thailand, the definition of fragility, or osteoporotic, hip fracture is a hip fracture resulting from low energy traumatic mechanisms in patients aged at least 50 years old [2]. Fragility hip fractures are often classified into two broad categories: femoral neck fracture and pertrochanteric femoral fracture (intertrochanteric or subtrochanteric femoral fracture). Operative treatment is generally preferred over conservative non-surgical management due to its significant superiority in reducing morbidity and mortality [3].

The primary treatment goal was to enable the patients to return to their pre-fracture state with optimal ambulation and the ability to self-care. Achieving the patients’ ability to ambulate independently can maximize their full functional potential and prevent serious complications, such as venous thromboembolism, pressure sores/ulcers, pneumonia, or urinary tract infections [4,5]. Moreover, mobility dependency was strongly associated with the survival rate of the patients after hip fractures. Although about one-third of patients with a previous history of hip fractures were able to return to their pre-fracture mobility status or functional independence, and up to two-thirds of these patients can complete activities of daily living (ADL) without difficulty [6], almost half of the patients who were dependent on essential activity and basic mobility died within five years of their injury [7]. Therefore, postoperative rehabilitation should be highly encouraged and incorporated into patient management plans as it is one of the keys to regaining function and preventing mobility dependency [8].

To properly deal with patient and their families’ concerns regarding ambulation and functional outcomes after surgery, the orthopedic surgeons or general physicians should understand the natural progression of the disease and prognostic factors that affect patients’ status. This study aimed to explore accurate pre-, intra-, and postoperative prognostic factors that affect the inability to bear self-weight at discharge in patients with fragility femoral neck fracture in the Thai clinical context.

## 2. Methodology

### 2.1. Study Design

This prognostic factor research was conducted with a retrospective cohort design. Electronic medical records of patients aged ≥ 50 years old diagnosed with a fragility femoral neck fracture between 1 January 2015 and 31 December 2019 were reviewed and retrieved. Patients with pathological fractures due to bone tumors or bone metastasis or patients who were referred to other hospitals were excluded from this study.

### 2.2. Data Collection

The data on potential prognostic factors were collected in three separate categories: pre-operative factors, intra-operative factors, and post-operative factors. 

For pre-operative factors, we collected the data on sex [9,10], age [9,11], body mass index (BMI) [12], pre-fracture ambulatory status (i.e., independent ambulation or walking without using gait aids, ambulation with gait aids (crutch, cane, or walker), ambulation in a wheelchair and non-ambulatory status [11,13,14,15], serum albumin level [16,17], associated fractures [18], second hip fracture [19], surgical techniques [20], and comorbidities, which might affect patients’ rehabilitation, walking ability or muscle strength [21,22].

For intra-operative factors, the data from operative notes and anesthetic records were extracted, such as the amount of time from admission to surgery, total anesthetic time, and the volume of intra-operative blood loss [23]. 

Post-operative factors included post-operative intensive care unit (ICU) admission or ventilator use, major post-operative complications (e.g., pulmonary embolism, intracerebral hemorrhage or ischemia, shock with any cause, myocardial infarction, cardiac arrest, or death) [24], other operations performed during admission, post-operative sedative drug use, pain score at initial rehabilitation [25], post-operative blood transfusion [26], urinary catheter use at post-operative day 2 [10], and pressure sore after operation [27]. 

### 2.3. Study Endpoint

The endpoint of interest is the ambulation status at discharge for each patient. All patients were classified as either unable or able to bear self-weight at discharge. Patients who were unable to bear self-weight were defined as patients who could only ambulate in a wheelchair or patients who could not ambulate (non-ambulatory status). In contrast, patients who were able to bear self-weight were defined as patients who had independent ambulation or patients who could ambulate with gait aids. 

### 2.4. Statistical Analysis

Statistical analysis was performed using Stata Statistical Package version 16.1 (StataCorp, College Station, TX, USA.). A two-sided *p*-value < 0.05 was considered statistically significant. Categorical variables were described by frequency and percentage. Based on data distribution, continuous variables were presented with mean and standard deviation or median and interquartile range (IQR). Fisher’s exact probability test was used to compare the differences in proportion across the two study groups (i.e., unable, or able to bear self-weight at discharge). Fisher’s exact was chosen over the chi-square test due to the relatively small number of sample size. By contrast, continuous variables were compared using Student’s *t*-test (parametric) or the Mann–Whitney U (non-parametric) test, depending on the data distribution. 

Univariable and multivariable risk ratio regression under Poisson distribution, or Poisson working model, were used to explore and identify associations between prognostic factors and an inability to bear self-weight at discharge. In performing statistical modeling, we executed three separate models: pre-operative factors model, intra-operative factors model, and post-operative factors model. Each model would be adjusted by the confounder summary score derived from other models. For instance, all pre-operative factors would be conditioned on intra-operative and post-operative confounder summary scores [28]. 

The multiple imputation with chained equation method (MICE) with a total of 20 imputed datasets was used to impute the missing values of factors with less than 50% missing. Sex, age, pre-fracture ambulation status, comorbidity, types of surgical treatments, and ambulation status at discharge were selected as independent variables to predict the mean of those missing values by using linear regression. Any variables with more than 50% missing data were not considered potential prognostic factors and, thus, were not included in the analysis.

## 3. Results

### 3.1. Patient Characteristics

Two-hundred and sixty-nine patients with fragility femoral neck fracture were included in this cohort. Of these patients, 55 (20.45%) were unable to bear their self-weight at discharge after surgery (Figure 1). Most of the patients were female (71.75%), with a mean age of 76.99 ± 10.06 years old. There were no significant differences between the two groups regarding sex, age, obesity, second hip fracture, and surgical techniques used. However, higher proportions of patients with underlying disease (i.e., ESRD, cirrhosis, cerebrovascular diseases, and psychiatric disorder/drug abuse), hypoalbuminemia, and associated fractures were observed in groups of patients who were unable to bear self-weight at discharge (Table 1). Pre-fracture ambulation status was significantly different between the two groups (*p* < 0.001). Missing data on the following prognostic variables was observed: BMI, serum albumin level, and median pain score on rehabilitation day. However, the proportion of missing data was less than 20% (Table 1).

For intra-operative factors, patients whose total time from admission to surgery was higher than 48 h and patients with longer anesthetic times were more likely to be unable to bear self-weight at discharge (Table 2). However, intra-operative blood loss was not significantly associated with the endpoint of interest. The proportions of patients with post-operative ICU admission or ventilator use, major post-operative complications, other operations performed during admission, postoperative sedative drug use, urinary catheter use on the second day after surgery, and the presence of pressure sore after operation were significantly higher in groups who were unable to bear self-weight (Table 2).

### 3.2. Prognostic Factors of the Inability to Bear Self-Weight at Discharge

The multivariable risk ratio regression results are shown separately for pre-operative, intra-operative, and post-operative factors in Table 3, Table 4 and Table 5, respectively. This study identified seven independent prognostic factors for the inability to bear self-weight at discharge in patients with fragile neck fractures: ESRD (RR = 2.29, CI = 1.03–5.10; *p* = 0.042), cirrhosis (RR = 3.16, CI = 1.48–6.76; *p* = 0.003), cerebrovascular diseases (RR = 2.68, CI = 1.32–5.43; *p* = 0.006), pre-fracture ambulation with gait aids (RR = 1.63, CI = 1.02–2.61; *p* = 0.040), non-ambulatory before the fracture event (RR = 11.18, CI = 5.86–21.32; *p* < 0.001), having associated fractures (RR = 3.75, CI = 1.99–7.07; *p* < 0.001), every 100 mL of intra-operative blood loss (RR = 1.11, CI = 1.03–1.19; *p* = 0.008), and having pressure sores after surgery (RR = 3.22, CI = 1.45–7.13; *p* = 0.004). 

## 4. Discussion

The geriatric- or fragility hip fracture is one of the health problems in every country. This study focuses on femoral neck fracture because the treatment of choices differs significantly from those of pertrochanteric fractures. In treating patients with fragility femoral neck fracture, especially for displaced fracture, arthroplasty is generally preferred over fixation [20,29,30]. By contrast, fracture fixation with intramedullary nailing is the mainstay in patients with pertrochanteric fractures [31]. During the post-operative period, patients who receive arthroplasty usually experience less pain and are able to ambulate and bear weight early [20]. Therefore, identifying prognostic factors in this specific group of patients is clinically meaningful and might be more accurate and specific than the factors identified in previous studies [32,33]. 

The ambulatory status at discharge is one of the issues to which the patients and their caregivers were concerned [34], as the worse the patient’s functional status was, the greater the burden would be for the caregivers [35]. Not only did it affect the quality of life of the caregivers, but ambulation status at discharge also was a known prognostic factor for survival at five years [7]. The incidence of patients who were unable to bear self-weight after surgery in our study was estimated at 20.45%, which was higher than the figures reported in previous studies, ranging from 10 to 16% [7,10]. The variation of this unfavorable event might be explained by the differences in patient characteristics and the study location. 

In this study, we have identified seven significant prognostic factors for the inability to bear self-weight at discharge in patients with fragility femoral neck fractures who underwent surgical operations: ESRD, cirrhosis, cerebrovascular disease, ambulation with gait aids or non-ambulatory status before the fracture event, having associated fractures, high intra-operative blood loss, and the presence of pressure sore after surgery. For clinical implementation, these factors could be classified into two groups, modifiable and non-modifiable factors. Patients’ comorbidities (i.e., ESRD, cirrhosis, and cerebrovascular disease) and pre-fracture ambulation status are non-modifiable. However, patients with these conditions should not be overlooked. Proper pre-operative management should be performed to ensure that the patients are fit for surgery. The presence of associated fractures, higher volume of intra-operative blood loss, and the presence of pressure sore after surgery are, in contrast, practically modifiable factors. 

Associated fractures occur in approximately 4 to 6.5% of patients diagnosed with femoral neck fractures [36]. Most associated fractures are upper limb fractures, mainly distal radial and proximal humeral fractures [36]. Compared to isolated hip fractures, patients with a concomitant upper extremity fracture had higher odds of death in the hospital (OR = 1.3. CI 1.2–1.4), were less likely to be discharged to home (OR = 0.73, CI 0.68–0.78), and had a significantly longer average length of stay (7.1 vs. 6.4 days, *p* < 0.01). Rehabilitation of patients diagnosed with hip fractures and associated fractures can be difficult and challenging due to pain, balance, and gait issues [18]. Orthopedists should be concerned about the stability and deliver rigid fixation for the associated fractures, which might facilitate early ambulation and improve the overall outcomes of the rehabilitation programs [37].

More intra-operative blood loss causes anemia, which is related to many unfavorable postoperative outcomes, especially delirium and cardiovascular event triggering. These events are significant barriers to the successful implementation of early rehabilitation programs among geriatric patients [38]. The proposed solution for this issue is adequate maintenance of hemoglobin (Hb) level [39,40], opting for minimally invasive surgery, and adequate intra-operative bleeding control [41]. The presence of pressure sores was a well-known indicator of inadequate quality of care [42,43], and was one of the major obstacles to postoperative rehabilitation [27]. Therefore, health care providers and caregivers should encourage and help the patient change the bed positioning, with early mobilization with gait aids or a wheelchair, and provide adequate wound care [43]. 

Non-modifiable factors should be controlled and corrected as much as possible. For example, adequate intake of calories could improve nutritional status among patients with chronic diseases [44,45]. Nonetheless, the rehabilitation program for patients with poor pre-fractural ambulation status is often more complex and extremely challenging for physiotherapists and caregivers because ambulation status significantly worsens after the fracture event [46,47,48]. Thus, modern concepts concerning the application of multidisciplinary care teams and specific care programs (e.g., increasing the number of visits by geriatricians, early geriatric rehabilitation program (EGR)) are keys to improving the surgical outcomes and patients’ quality of life. Previous studies had reported the outcomes of implementing these programs, which results in a significant increase in walking ability, early initiation of anti-osteoporotic agents for preventing secondary hip fractures, and decreased patient mortality [49,50].

There were some limitations to this study. Firstly, this study is conducted using a retrospective cohort study design. The presence of missing data on prognostic variables that were not routinely collected has the potential to bias our results. Fortunately, less than 20% of missing data were identified. Moreover, standard imputation methods were used to account for this issue properly. Secondly, only relatively small samples were available to be included compared to the number of potential predictors pre-specified to be analyzed. According to power back calculation, our current sample size was inadequate to identify statistically significant differences of several predictors within the model (Appendix A). However, the inability to achieve a sufficient sample size to identify statistical significance should not discourage observational data analysis, as the goal is not to detect significance but rather to quantify the existing association in question [51]. Moreover, the direction and magnitude of the identified effect estimates, which better reflect the clinical significance of the associations, should be the primary focus rather than the statistical significance based on *p*-values, which were sample size-dependent. Finally, the results of our study might be generalizable to health care settings with similar clinical contexts and patient or healthcare characteristics. External validation of our results in other countries, or different settings, should be warranted.

## 5. Conclusions

The ability to bear self-weight is one of the expectations of physicians, patients with fragility hip fractures, and their families. Significant prognostic factors that affect the inability to bear self-weight at discharge were ESRD, cirrhosis, cerebrovascular diseases, pre-fracture ambulatory status, associated fractures, increased intra-operative blood loss, and the occurrence of pressure sores after surgery. Prognostic factors are not used to judge the patients’ status at discharge nor to decide whether surgical operations should be performed but are used as essential prognostic information to communicate with the patients and their families during shared decision-making. Multidisciplinary collaboration with holistic care during the perioperative period is key to improving patients’ outcomes and quality of life. 

## Figures and Tables

**Figure 1 ijerph-19-03992-f001:**
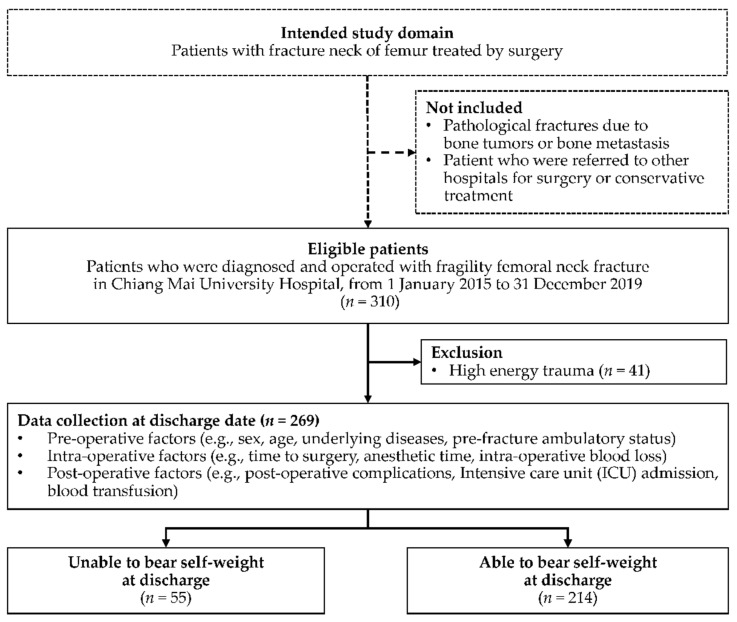
Study flow diagram of patients who were operated on due to fragility femoral neck fracture during the study period and the proportion of patients who were unable and able to bear self-weight at discharge.

**Table 1 ijerph-19-03992-t001:** Pre-operative factors of patients with fragility femoral neck fracture between patients who were unable and able to bear self-weight at discharge.

	Missing Value *n* (%)	Unable to Bear Self-Weight at Discharge (*n* = 55) *n* (%)	Able to Bear Self-Weight at Discharge (*n* = 214) *n* (%)	*p*-Value
Sex	0 (0)			
- Male		19 (34.55)	57 (26.64)	0.245
- Female		36 (65.45)	157 (73.36)	
Age ≥ 80 years	0 (0)	27 (49.09)	100 (46.73)	0.764
BMI ≥ 25 kg/m^2^	3 (1.12)	8 (14.81)	41 (19.34)	0.557
Underlying diseases				
- ESRD ^a^	0 (0)	11 (20.00)	13 (6.07)	0.003
- Cirrhosis	0 (0)	3 (5.45)	0 (0)	0.008
- Cerebrovascular diseases	0 (0)	10 (18.18)	17 (7.94)	0.040
- Psychiatric disorders/Drug abuse	0 (0)	6 (10.91)	5 (2.34)	0.011
- Parkinson disease	0 (0)	2 (3.64)	6 (2.80)	0.668
- Diabetes mellitus	0 (0)	12 (21.82)	48 (22.43)	1.000
- Heart diseases	0 (0)	12 (21.82)	36 (16.82)	0.430
- COPD ^b^/Asthma	0 (0)	5 (9.09)	15 (7.01)	0.571
- Eye diseases ^c^	0 (0)	4 (7.27)	16 (7.48)	1.000
- Cancer	0 (0)	3 (5.45)	16 (7.48)	0.773
- Dementia	0 (0)	5 (9.09)	19 (8.88)	1.000
Pre-fracture ambulation status				
- Independent ambulation	0 (0)	29 (52.73)	166 (77.57)	<0.001
- Ambulation with gait aids	0 (0)	22 (40.00)	48 (22.43)	
- Ambulation in wheelchair	0 (0)	2 (3.64)	0 (0)	
- Non-ambulatory status	0 (0)	2 (3.64)	0 (0)	
Hypoalbuminemia (<3.5 g/dl)	45 (16.73)	23 (50.00)	48 (26.97)	0.004
Associated fractures	0 (0)	10 (18.18)	2 (0.93)	<0.001
Second hip fracture	0 (0)	6 (10.91)	20 (9.35)	0.798
Surgical techniques				
- Arthroplasty ^d^	0 (0)	41 (74.55)	165 (77.10)	0.722
- Fixation ^e^	0 (0)	14 (25.45)	49 (22.90)	

^a^ ESRD = end stage renal disease; ^b^ COPD = chronic obstructive pulmonary disease; ^c^ Eye diseases include blinded, cataract and glaucoma; ^d^ Arthroplasty include total hip arthroplasty and hemiarthroplasty; ^e^ Fixation includes multiple screw fixation and dynamic hip screw fixation (DHS).

**Table 2 ijerph-19-03992-t002:** Intra-operative and post-operative factors of patients with fragility femoral neck fracture between patients who were unable and able to bear self-weight at discharge.

	Missing Value *n* (%)	Unable to Bear Self-Weight at Discharge (*n* = 55) *n* (%)	Able to Bear Self-Weight at Discharge (*n* = 214) *n* (%)	*p*-Value
**Intra-operative factors**				
Delayed surgery (time from admission to surgery > 48 h)	0 (0)	51 (92.73)	174 (81.31)	0.042
Anesthetic time (hours) ^a^	0 (0)	2.17 (1.92, 2.50)	2.00 (1.75, 2.25)	0.010 ‡
Intra-operative blood loss (ml) ^a^	0 (0)	100 (50, 200)	100 (90, 200)	0.712 ‡
**Post-operative factors**				
Post-operative ICU admission or ventilator use	0 (0)	10 (18.18)	4 (1.87)	<0.001
Major post-operative complications	0 (0)	9 (16.36)	5 (2.34)	<0.001
Other operation in admission	0 (0)	5 (9.09)	4 (1.87)	0.020
Post-operative sedative drug use	0 (0)	28 (50.91)	58 (27.10)	0.001
Post-operative blood transfusion	0 (0)	18 (32.73)	50 (23.36)	0.166
Urinary catheter use at post-operative day 2	0 (0)	22 (40.00)	38 (17.76)	0.001
Moderate to severe pain score at rehabilitation day (PS = 4–10)	30 (5.78)	7 (14.29)	30 (14.56)	1.000
Pressure sore	0 (0)	3 (5.45)	1 (0.47)	0.028

‡ Mann–Whitney U test, ^a^ Median (IQR).

**Table 3 ijerph-19-03992-t003:** Univariable and multivariable risk ratio of pre-operative prognostic factors of the inability to bear self-weight at discharge in patients with fragility femoral neck fracture.

	Univariable RR	95% CI	*p*-Value	Multivariable RR *	95% CI	*p*-Value
Male	1.34	0.77–2.34	0.302	1.49	0.83–2.68	0.185
Age ≥ 80 years	1.08	0.64–1.83	0.280	0.93	0.53–1.62	0.793
BMI ≥ 25 kg/m^2^	0.80	0.38–1.70	0.568	1.02	0.49–2.10	0.956
Comorbidity						
- ESRD ^a^	2.55	1.32–4.94	0.005	2.29	1.03–5.10	0.042
- Cirrhosis	5.12	1.60–16.38	0.006	3.16	1.48–6.76	0.003
- Cerebrovascular diseases	1.99	1.00–3.95	0.049	2.68	1.32–5.43	0.006
- Psychiatric disorders/Drug abuse	2.87	1.23–6.70	0.015	2.02	0.93–4.35	0.074
- Parkinson disease	0.98	0.27–3.55	0.977	1.06	0.48–2.34	0.880
- Diabetes mellitus	0.97	0.51–1.84	0.931	1.13	0.63–2.02	0.691
- Heart diseases	1.28	0.68–2.44	0.443	0.90	0.49–1.63	0.721
- COPD ^b^/Asthma	1.25	0.50–3.12	0.640	1.36	0.57–3.21	0.489
- Eye diseases ^c^	0.98	0.35–2.70	0.963	1.35	0.59–3.06	0.480
- Cancer	0.76	0.24–2.43	0.643	0.85	0.26–2.78	0.783
- Dementia	1.02	0.41–2.56	0.965	1.27	0.52–3.09	0.600
Pre-fracture ambulation status						
- Independent ambulation	Ref.					
- Ambulation with gait aids	2.11	1.21–3.68	0.008	1.63	1.02–2.61	0.040
- Ambulation in wheelchair	6.72	1.60–28.18	0.009	4.45	0.96–20.65	0.056
- Non-ambulatory status	6.72	1.60–28.18	0.009	11.18	5.86–21.32	<0.001
Hypoalbuminemia	2.11	1.24–3.58	0.006	1.61	0.92–2.79	0.094
Associated fractures	4.76	2.40–9.44	<0.001	3.75	1.99–7.07	<0.001
Second hip fracture	1.14	0.49–2.67	0.755	1.09	0.59–2.02	0.784
Fixation surgery	1.12	0.61–2.05	0.722	1.66	0.94–2.93	0.081

Risk ratio regression with Poisson working model; * The model was adjusted with intra- and post-operative confounder summary score; ^a^ ESRD = end stage renal disease; ^b^ COPD = chronic obstructive pulmonary disease; ^c^ Eye diseases include blinded, cataracts and glaucoma.

**Table 4 ijerph-19-03992-t004:** Univariable and multivariable risk ratio of intra-operative prognostic factors of the inability to bear self-weight at discharge in patients with fragility femoral neck fracture.

	Univariable RR	95% CI	*p*-Value	Multivariable RR *	95% CI	*p*-Value
Delayed surgery (time from admission to surgery > 48 h)	2.49	0.90–6.90	0.078	1.63	0.61–4.34	0.325
Anesthetic time (every 1 h)	1.47	1.02–2.12	0.037	0.87	0.66–1.15	0.332
Intra-operative blood loss (every 100 mL)	1.13	0.97–1.31	0.108	1.11	1.03–1.19	0.008

Risk ratio regression with Poisson working model; * The model was adjusted with pre- and post-operative confounder summary score.

**Table 5 ijerph-19-03992-t005:** Univariable and multivariable risk ratio of post-operative prognostic factors of the inability to bear self-weight at discharge in patients with fragility femoral neck fracture.

	Univariable RR	95% CI	*p*-Value	Multivariable RR *	95% CI	*p*-Value
Post-operative ICU admission or ventilator use	4.05	2.04–8.03	<0.001	1.72	0.76–3.85	0.191
Major post-operative complications	3.56	1.74–7.28	<0.001	1.06	0.45–2.52	0.896
Other operation in admission	2.89	1.15–7.24	0.024	0.62	0.26–1.46	0.272
Post-operative sedative drug used	2.21	1.30–3.74	0.003	1.19	0.68–2.10	0.547
Post-operative blood transfusion	1.44	0.82–2.53	0.206	1.26	0.77–2.04	0.357
Urinary catheter use at post-operative day 2	2.32	1.35–3.98	0.002	1.42	0.91–2.21	0.126
Moderate to severe pain at rehabilitation day	0.86	0.34–2.16	0.749	1.31	0.67–2.55	0.435
Pressure sore	3.82	1.19–12.24	0.024	3.22	1.45–7.13	0.004

Risk ratio regression with Poisson working model; * The model was adjusted with pre- and intra-operative confounder summary score.

## Data Availability

Not applicable.

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
