# Peer review of "Prognostic Factors of the Inability to Bear Self-Weight at Discharge in Patients with Fragility Femoral Neck Fracture: A 5-Year Retrospective Cohort Study in Thailand"

_ijerph, 2022, doi:10.3390/ijerph19073992_

Round 1

Reviewer 1 Report

THank you for the opportunity to reveiw the manuscript. Overall it needs to flow better. 

Author Response

Dear Editor and reviewers,

            We would like to thank you for the opportunity to revise our manuscript to be qualified for publication in the International Journal of Environmental Research and Public Health. We have revised and modified most parts of our manuscript as addressed in response to reviewers’ comments. We hope that our responses and revisions would substantially improve the quality of our manuscript and would be qualified for publication in the journal. If there were any further questions or minor points to be addressed or elaborated on, please let us know. We would be more than eager to make any further revisions.

Reviewer 2 Report

This study explores pre-operative, intra-operative, and postoperative prognostic factors that affect ambulation at discharge time in geriatric patients with fragile femoral neck fractures.
The group with the Ability to self-weight bearing at discharge is four times bigger than the one unable to do so. The statistical analysis does not account for this size effect, and there are many inconsistencies. The paper needs significant improvement in scientific and English writing (for instance, the discussion is hard to follow and too long). 

Major comments:

There is a problem with some groups where the number of patients is pretty low ( under 6 ), and the test performed was not a nonparametric test.
For groups above 6, please test the distribution and use parametric or nonparametric tests accordingly.
For groups under 6, please use a nonparametric test.
Calculate the error possible, power, and the number of samples needed to reach significance. 
Provide supplemental files and add them to the discussion when appropriate.
Then, based on the new results, please re-arrange your discussion appropriately. 

The authors could discuss more their results in regards of :

Gleich J, Fleischhacker E, Rascher K, Friess T, Kammerlander C, Böcker W, Bücking B, Liener U, Drey M, Höfer C, Neuerburg C. Increased Geriatric Treatment Frequency Improves Mobility and Secondary Fracture Prevention in Older Adult Hip Fracture Patients-An Observational Cohort Study of 23,828 Patients from the Registry for Geriatric Trauma (ATR-DGU). J Clin Med. 2021 Nov 23;10(23):5489. doi: 10.3390/jcm10235489. PMID: 34884190; PMCID: PMC8658325.

Schoeneberg C, Pass B, Volland R, Knobe M, Eschbach D, Ketter V, Lendemans S, Aigner R; Registry for Geriatric Trauma DGU. Four-month outcome after proximal femur fractures and influence of early geriatric rehabilitation: data from the German Centres of Geriatric Trauma DGU. Arch Osteoporos. 2021 Apr 12;16(1):68. doi: 10.1007/s11657-021-00930-9. PMID: 33846869.

Minor comments
Figure 1 has the tabs appearing. 

Author Response

(The authors gave the same response as above.)
